# Prescribing the aerosol effective radiative forcing in the Simple Cloud-Resolving E3SM Atmosphere Model v1

Naser Mahfouz<sup>1</sup>, Hassan Beydoun<sup>2</sup>, Johannes Mülmenstädt<sup>1</sup>, Noel Keen<sup>3</sup>, Adam C. Varble<sup>1</sup>, Luca Bertagna<sup>4</sup>, Peter Bogenschutz<sup>2</sup>, Andrew Bradley<sup>4</sup>, Matthew W. Christensen<sup>1</sup>, T. Conrad Clevenger<sup>4</sup>, Aaron Donahue<sup>2</sup>, Jerome Fast<sup>1</sup>, James Foucar<sup>4</sup>, Jean-Christophe Golaz<sup>2</sup>, Oksana Guba<sup>4</sup>, Walter Hannah<sup>2</sup>, Benjamin Hillman<sup>4</sup>, Robert Jacob<sup>5</sup>, Wuyin Lin<sup>6</sup>, Po-Lun Ma<sup>1</sup>, Yun Qian<sup>1</sup>, Balwinder Singh<sup>1</sup>, Christopher Terai<sup>2</sup>, Hailong Wang<sup>1</sup>, Mingxuan Wu<sup>1</sup>, Kai Zhang<sup>1</sup>, Andrew Gettelman<sup>1</sup>, Mark Taylor<sup>4</sup>, L. Ruby Leung<sup>1</sup>, Peter Caldwell<sup>2</sup>, and Susannah Burrows<sup>1</sup>

Correspondence: Naser Mahfouz (naser.mahfouz@pnnl.gov) and Susannah Burrows (susannah.burrows@pnnl.gov)

Abstract. Aerosol effective radiative forcing critically influences climate projections but remains poorly constrained. Using the Simple Cloud-Resolving E3SM Atmosphere Model (SCREAM) v1 configuration, we assess the sensitivity of the radiative forcing due to anthropogenic aerosol changes using a simplified prescribed aerosol scheme (SPA) derived from E3SM v3. Nudged simulations at 3 km and 12 km horizontal grid spacings reveal a more negative aerosol forcing than the reference 100-km E3SM v3 whence the SPA properties are derived. The resulting globally averaged aerosol forcing signal is largely due to aerosol—cloud interactions and exhibits little overall resolution sensitivity, while hints of resolution sensitivity appear regionally between the 3-km and 12-km runs. While the default SPA scheme overestimates cloud droplet dependence on aerosols, parameterization adjustments in the activation process reconcile forcing estimates with the reference model. Our results demonstrate the ability to use a prescribed aerosol scheme to hold aerosol forcing to a desired strength across resolutions.

# 10 1 Introduction

Atmospheric aerosol particles are an essential component of climate models (Intergovernmental Panel On Climate Change, 2014; Carslaw, 2022). Aerosols are known to influence the atmosphere state directly by altering its radiation budget through aerosol–radiation interactions and indirectly by altering cloud properties through aerosol–cloud interactions (Bellouin et al., 2020, and references therein). Through their role as cloud condensation nuclei and ice nuclei, aerosol particles modify liquid and ice clouds (e.g., Mülmenstädt et al., 2019; Burrows et al., 2022). Despite significant progress in aerosol-related studies over recent decades, many questions remain about the role of aerosols in climate projections — particularly those involving aerosol–cloud interactions (Kreidenweis et al., 2019).

<sup>&</sup>lt;sup>1</sup>Pacific Northwest National Laboratory, Richland, WA, USA

<sup>&</sup>lt;sup>2</sup>Lawrence Livermore National Laboratory, Livermore, CA, USA

<sup>&</sup>lt;sup>3</sup>Lawrence Berkeley National Laboratory, Berkeley, CA, USA

<sup>&</sup>lt;sup>4</sup>Sandia National Laboratories, Albuquerque, NM, USA

<sup>&</sup>lt;sup>5</sup>Argonne National Laboratory, Lemont, IL, USA

<sup>&</sup>lt;sup>6</sup>Brookhaven National Laboratory, Upton, NY, USA

Accurately modeling aerosols and their effects on the atmosphere is complicated by their variability in source, composition, and size (e.g., Seinfeld and Pandis, 2016; Topping and Bane, 2022). Additional complexity arises in processes they undergo and processes they influence (Carslaw, 2022). Aerosols can be emitted into the atmosphere as fully formed particles from natural and anthropogenic sources (Seinfeld and Pandis, 2016), or they can be formed in the atmosphere from precursor gases via gas-to-particle conversion and new-particle formation that have their origin in natural or anthropogenic sources (Dunne et al., 2016). They can undergo a variety of processes in the atmosphere, such as growing or shrinking via condensation or evaporation as well as coagulation growth or loss via collisions with other particles (Fuchs, 1964; Fuchs and Sutugin, 1970; Seinfeld and Pandis, 2016). They can additionally be scavenged by clouds and precipitation, sometimes reemerging by resuspension, and they can be removed from the atmosphere by wet and dry deposition (Adams and Seinfeld, 2002; Wang et al., 2020). Whether direct or indirect, aerosol effects on the atmosphere and thus climate depend closely on the aforementioned processing as well as their concentration, composition, and size (Adams et al., 2013; Carslaw, 2022).

Due to the complexity associated with aerosols, they are generally represented through simplified schemes in atmosphere models, varying from prescribed (e.g., Stevens et al., 2017) to modal (e.g., Stier et al., 2005; Liu et al., 2012) and interactive sectional (e.g., Tilmes et al., 2023) schemes. In all, the multi-scale complexity of aerosol representation is usually reduced to balance process realism with computational performance depending on the need. Notwithstanding, fully constraining aerosol effects on the atmosphere remains elusive. Notably, definitively constraining the effective aerosol forcing to anthropogenic aerosol perturbation remains mired in uncertainty (Intergovernmental Panel On Climate Change, 2023). One of the main challenges is understanding the adjustments (Quaas et al., 2024) to aerosol perturbations. While the instantaneous radiative response to an aerosol change is relatively well understood (Twomey effect; Twomey, 1977; Quaas et al., 2020), subsequent adjustment processes like precipitation suppression (Albrecht, 1989) and entrainment feedback (Ackerman et al., 2004; Bretherton et al., 2007) lead to changes in macroscopic cloud properties (like cloud water amount and cloud fraction) that are far less constrained (Gryspeerdt et al., 2019; Bellouin et al., 2020).

With the dawn of new high-resolution global atmosphere models, it is hoped that some of the uncertainty in aerosol-cloud interactions can be reduced (e.g., Sato et al., 2018; Terai et al., 2020; Mülmenstädt and Wilcox, 2021; Herbert et al., 2024; Weiss et al., 2024). Current state-of-the-art climate models, which typically have an effective resolution of about 100 km, tend to have interactive aerosol schemes. On the other hand, next-generation high-resolution models running with an effective resolution below 10 km are computationally expensive, and so they often use the simplest and cheapest representation for aerosols. Most high-resolution models use prescribed aerosol schemes but some are being developed with interactive aerosol schemes (Weiss et al., 2024, and references therein). Even with prescribed aerosols, high-resolution models can still provide valuable insights into aerosol-cloud interactions (e.g., Herbert et al., 2024).

40

In this study, we examine the sensitivity of the Energy Exascale Earth System Model (E3SM) Simple Cloud-Resolving E3SM Atmosphere Model (SCREAM) v1 configuration to aerosol changes during the industrial era. Our focus is the global quantification of the aerosol effective radiative forcing (ERFaer), and especially its dominant component: the aerosol–cloud interactions. Goto et al. (2020) report three-year-long global simulations at a 14-km horizontal grid spacing with an interactive aerosol scheme, including sensitivity studies to anthropogenic aerosols. They show that most features related to aerosols

(especially primary aerosols) improve slightly with finer grid spacing, but the ERFaer sensitivity to resolution is small (Goto et al., 2020). Additionally, Weiss et al. (2024) report a one-year-long global simulation at a 5-km horizontal grid spacing with interactive modal aerosols at present-day conditions using a one-moment scheme that predicts number concentration with other aerosol properties prescribed. They highlight the emergence of mesoscale features of natural aerosols like dust aerosol patterns in desert storms and the interplay of sea salt aerosol with tropical cyclones (Weiss et al., 2024). Both studies utilized free-running simulations that are not nudged to reanalysis data. Our study differs from both in its exclusive focus on the global ERFaer signal obtained from a prescribed aerosol scheme, using pairs of one-year-long simulations at 3-km, 12-km, and 100-km effective resolutions, all nudged to reanalysis data in order to deduce a robust ERFaer signal. While an interactive treatment of aerosols offers more realism overall, it is also more computationally expensive and less flexible for disentangling process pathways due to the two-way nature of the coupling between aerosols and the model state. In the following manuscript, we focus on the aerosol activation process and highlight its importance in determining the ERFaer signal. We note that an interactive aerosol scheme is in development for the SCREAM configuration and should be available in the near term.

# 65 2 Methods

# 2.1 Modeling framework

# 2.1.1 SCREAM v1

We use the Energy Exascale Earth System Model (E3SM) Simple Cloud-Resolving E3SM Atmosphere Model (SCREAM) v1 configuration in this study. The SCREAM v1 configuration is described in detail by Donahue et al. (2024), and we adopt changes outlined in their Section 5.5. We briefly outline parts of the configuration most pertinent to this study. The simplified higher-order closure scheme (SHOC; Bogenschutz and Krueger, 2013) is used to parameterize cloud macrophysics and turbulent vertical mixing. A modified version of the predicted particle properties scheme (P3; Morrison and Milbrandt, 2015) is used for cloud microphysics parameterization. In the modified P3 version, there exists only one ice category and the liquid supersaturation is removed for consistency with SHOC's liquid saturation adjustment assumption (Caldwell et al., 2021). Additionally, the ice cloud fraction is set based on a grid-mean ice mass mixing ratio threshold — all or nothing based on the condition:  $q_i > 10^{-14}$  kg kg<sup>-1</sup> (Caldwell et al., 2021). The liquid cloud fraction is continuous and provided by SHOC.

Aerosol effects on the model physics are parameterized by the simple prescribed aerosol (SPA) scheme. The SPA scheme prescribes the optical properties of aerosols, so that they are accounted for in the radiation scheme (RRTMGP; Pincus et al., 2019). The SPA aerosol optical properties are the three-dimensionally resolved and band-resolved single scattering albedo, asymmetry parameter, as well as the shortwave and longwave optical depths. Furthermore, SPA is linked to the model microphysics via its impact on the prognostic liquid cloud droplet number concentration,  $N_c$ . SPA prescribes a three-dimensional grid-mean cloud condensation nuclei concentration,  $N_{ccn}$ , whose value scaled by the three-dimensional liquid cloud fraction ( $C_f$ ) is used to update grid-mean  $N_c$  according to Equation 1.

$$N_{\rm c} = \max\left(N_{\rm c}, \alpha\left(N_{\rm ccn}C_{\rm f}\right)\right) \tag{1}$$

The function  $\alpha$  ( $N_{\rm ccn}C_{\rm f}$ ) represents the activation process of cloud condensation nuclei into liquid cloud droplets. For simplicity, the default SCREAM v1 configuration assumes a linear function of activation  $\alpha$ , defined as  $\alpha$  ( $N_{\rm ccn}C_{\rm f}$ ) =  $N_{\rm ccn}C_{\rm f}$ . This represents the upper limit of theoretically possible values (e.g., Chen et al., 2011, Equation 5), though some global models show values exceeding that (e.g., Gryspeerdt et al., 2023, Section 2.2). By default, SPA uses  $N_{\rm ccn}$  at 0.10 % supersaturation as a reasonable proxy for cloud condensation nuclei concentration and for its availability as an output and as mean representative for all cloud conditions. For the rest of the manuscript, unless explicitly stated otherwise, we use  $N_{\rm ccn}$  as a shorthand for cloud condensation nuclei concentration at 0.10 % supersaturation. Both the optical properties and  $N_{\rm ccn}$  are prescribed as climatological monthly in-grid means at a coarser resolution than the model's (see Section 2.1.2); during the simulation, the values are interpolated to the model's resolution and time step. The P3 ice nucleation process is not aerosol-aware, and so it is not directly affected by the SPA scheme (Donahue et al., 2024).

Although generally similar to the MACv2-SP scheme (Stevens et al., 2017) and variations thereof (Fiedler et al., 2017; Herbert et al., 2024), the SPA scheme in SCREAM v1 has some key differences. First, even though MACv2-SP is derived from an observational dataset, it still requires a reference  $N_c$  (i.e., a simulated preindustrial background  $N_c$ ) obtained from a model run to estimate anthropogenic aerosol changes (Stevens et al., 2017, Section 3). On the other hand, SPA uses a more consistent approach by prescribing all aerosol optical and cloud properties from a previous model run. Second, while the MACv2-SP scheme prescribes only shortwave aerosol optical properties (Fiedler et al., 2017), the SPA scheme prescribes both shortwave and longwave aerosol optical depths. Third and most substantially, the MACv2-SP and SPA schemes differ significantly in how they are coupled to cloud microphysics. The original MACv2-SP accounts for the Twomey effect and its associated radiative adjustments but is not coupled to cloud microphysics (Fiedler et al., 2017, Section 2.2.1). A modified version of MACv2-SP (Herbert et al., 2024) includes further coupling to cloud microphysics but only through changing the autoconversion rate of cloud droplets to raindrops (Herbert et al., 2024, Section 2.1). In SCREAM v1, SPA is coupled to the cloud microphysics scheme directly by setting a climatological minimum cloud droplet number concentration,  $N_c$ . The SPA-derived minimum replenishes the prognostic  $N_c$  at each time step right before the cloud microphysics calculations. This ensures that aerosol effects are accounted for in the prognostic tracer  $N_c$ , while allowing for cloud processes in P3, vertical mixing in SHOC, and large-scale transport to modify it (Donahue et al., 2024, Section 2.1). We note that SPA offers the only source process for the prognostic N<sub>c</sub>. All other P3 cloud processes are sinks; they are autoconversion, accretion, sedimentation, riming, and freezing (Morrison and Milbrandt, 2015, Appendices B and C).

In summary, SPA replaces the "aerosol state" (e.g., Karset et al., 2020, "aerosol physics" in Figure 2) with a climatological mean state derived from long model runs with interactive aerosols. In doing so, it prescribes aerosol–radiation interactions through aerosol optical properties and sets a climatological minimum  $N_c$  to account for aerosol–cloud interactions. This represents a one-way coupling between the aerosol state and the rest of the model, wherein the aerosol state influences the model's state but not vice versa. Of special note, this one-way coupling allows for the study of the instantaneous effect of aerosol perturbation as well as adjustments to it, without the complications of feedbacks from the new cloud state to the aerosol state.

# 2.1.2 E3SM v3

The Energy Exascale Earth System Model (E3SM) v3 is a milestone release (E3SM Project, 2024), with significant improvements in the model's ability to reproduce the historical temperature record (Xie et al., 2025; Golaz et al., 2025). We use the 120 state-of-the-art E3SM v3 coarse-resolution atmosphere model, which is run on an unstructured grid whose effective horizontal resolution is 100 km with output at 150 km near the equator and 80 vertical levels. We briefly describe features of the E3SM v3 atmosphere model most pertinent to the study at hand. Unlike SCREAM v1, it uses the Cloud Layers Unified By Binomials scheme (CLUBB; Golaz et al., 2002) for the macrophysics parameterization of clouds and turbulent vertical mixing. 125 Similar to SCREAM v1, it uses the P3 scheme for stratiform cloud microphysics parameterization (Terai et al., 2024), but with slightly different tuning parameters (Shan et al., 2024) including a default  $N_c \ge 20 \text{ cm}^{-3}$  limiter. While SCREAM v1 has no parameterized deep convection, E3SM v3 uses a Zhang and McFarlane (1995) type convection scheme that also includes cloud microphysics effects that are modulated by interactive aerosols (Terai et al., 2024). E3SM v3 also uses the RRTMG radiation scheme (Clough et al., 2005). Most importantly, it has an interactive modal aerosol model (MAM) scheme, based on 130 MAM4 in previous E3SM versions (Wang et al., 2020), with new features that are being characterized and evaluated in detail for upcoming publications (e.g., Xie et al., 2025; Golaz et al., 2025).

We use the default E3SM v3 configuration in this study to produce the reference climatological mean state for SPA as described in Section 2.2.1. We also use the default E3SM v3 configuration to conduct additional sensitivity studies detailed in Section 2.2.2. Furthermore, we reconfigure E3SM v3 to prescribe aerosol effects using SPA as described in Section 2.1.1. We call the default E3SM v3 configuration E3SM-MAM and we call the modified E3SM v3 configuration with prescribed aerosols E3SM-SPA. The only differences between E3SM-MAM and E3SM-SPA are the nature of aerosol effects (interactive vs. prescribed, respectively) and the radiation scheme (RRTMG vs. RRTMGP, respectively), with minor differences in coupling interfaces between components.

### 2.2 Simulation protocol

# 140 2.2.1 Reference simulation

To reduce potential sampling biases in the SPA climatological means, we run the E3SM-MAM v3 model for 31 years with only the atmosphere and land components with prescribed sea ice extent and sea surface temperature. For the present-day simulation, all initial conditions and prescribed settings correspond to a climatological state representative of around 2010. For the pre-industrial run, we keep everything the same as the present-day run, but change the settings for aerosols and their precursors to a pre-industrial state in 1850. The simulations are free-running, that is, without the use of nudging to control the meteorological state. We discard the first year and obtain climatological monthly means for the last 30 years of the SPA fields of interest described in Section 2.1. The assessment of the E3SM-MAM v3 model is the subject of other pending manuscripts (e.g., Xie et al., 2025; Golaz et al., 2025), and as such we do not attempt to evaluate its performance here. We briefly note that its anthropogenic aerosol effective radiative forcing is shown in Figure 2 as  $-0.74 \text{ W m}^{-2}$  (more details are provided by Xie et al., 2025) and its aerosol vertically summed optical depth at 550 nm is shown in Figure 1 with an anthropogenic signal of

**Figure 1.** The vertically summed aerosol optical depth at 550 nm for the present-day (top) and pre-industrial (bottom) scenarios. Comparing the two scenarios, we see a clear anthropogenic aerosol signal over China and India, and to lesser extent over South America and western Africa.

approximately 0.03. In the present-day scenario, the diagnostic optical depth in Figure 1 indicates strong aerosol signal over eastern China, India, and western Africa as well as dust regions in the Middle and north Africa while a moderate signal is present in South America. On the other hand, in the pre-industrial scenario, the naturally occurring signal in the dust regions remains as well as a weak signal elsewhere. We note that the choice of E3SM-MAM v3 as the reference data for SPA implies that relevant biases in the reference low-resolution model will be in the new high-resolution model.

# 2.2.2 Sensitivity simulations






The main goal of this study is to examine the sensitivity of the SCREAM v1 configuration to aerosol perturbation, namely from pre-industrial to present-day conditions. To do so, we conduct two SCREAM v1 simulations: one with SPA pre-industrial aerosol conditions and another with SPA present-day aerosol conditions. With the SPA input files obtained from the reference simulation (Section 2.2.1), we run SCREAM v1 at two horizontal grid spacings (3 km and 12 km) for 13 months starting on August 1, 2019 and ending on September 1, 2020. This time period was selected based on the availability of boundary conditions as part of a series for sensitivity studies of the SCREAM v1 configuration. Specifically, a recent 13-month period was chosen with a weak El Niño Southern Oscillation starting from a northern hemisphere summer for a parallel radiative feedback study by Terai et al. (2025). The sea surface temperature is obtained from the Operational Sea Surface Temperature and Ice Analysis (Donlon et al., 2012; UKMO, 2012) and the sea ice coverage is obtained from the The European Organization for the Exploitation of Meteorological Satellite Ocean and Sea Ice Satellite Application Facility (UKMO, 2012). The atmosphere initial condition is obtained from the fifth-generation European Centre for Medium-Range Weather Forecasts atmospheric reanalyses (ERA5) data at a horizontal resolution of 0.25°×0.25° and native vertical level (Hersbach et al., 2020). The atmosphere initial condition is then interpolated to SCREAM v1's horizontal and vertical levels. The land initial condition is obtained from running the land model forced by atmospheric reanalysis from CRUNCEP data (Viovy, 2018) until the SCREAM v1 simulation start date (for 20 years for the 3-km runs and for 28 years for the 12-km runs).

To enhance the signal-to-noise ratio, we nudge the model horizontal winds towards the Modern-Era Retrospective Analysis for Research and Applications, version 2 (MERRA-2) reanalysis data (Gelaro et al., 2017; Global Modeling And Assimilation Office, 2015). The MERRA-2 data is obtained at a temporal resolution of six hours, horizontal resolution of 0.5°×0.625°, and native vertical levels (Global Modeling And Assimilation Office, 2015). The MERRA-2 data is then interpolated offline to an unstructured grid whose horizontal resolution is 100 km at the equator. During SCREAM v1 runs, the nudging data is interpolated online to the model's temporal, horizontal, and vertical levels and applied to the model's horizontal winds. The application of nudging is consistent with best practices established in the E3SM Project to diagnose aerosol forcing while controlling for variability due to large-scale circulation (e.g., Zhang et al., 2022a, b; Mahfouz et al., 2024). Specifically, the nudging data is read at a six-hourly cadence; it is temporally interpolated linearly in between analysis time-steps. Additionally, the nudging is applied with a relaxation timescale of six hours to the horizontal winds. We note that unlike Zhang et al. (2022a) who avoid nudging the lowest layers near the surface, we apply nudging in all vertical levels as we postulate the latter offers a more consistent approach with all the different configurations we have in this study (interactive aerosols or not; higher and lower resolutions).

We repeat the above procedure for the E3SM-MAM v3 and E3SM-SPA v3 configurations at 100 km, with the following difference: the model is run for 13 months starting on December 1, 2009, and nudged with MERRA-2 data corresponding to that period. We note that the source of the nudging data (MERRA-2 vs. ERA5) and the period of interest (2019–2020 like in SCREAM v1 runs vs. 2009–2010 in E3SM v3 runs) have a negligible effect on the ERFaer signal, and the choice of nudging data and period was made purely for practical reasons of using readily available, existing data. The use of nudging is solely

**Table 1.** Summary of simulations conducted in this study, labeled as they appear in the figures. Because the simulations are used to calculate ERFaer, the simulations appear in pairs. For each pair, the only difference between the two simulations is the aerosol state, either preindustrial or present-day. The simulations are described in detail in Section 2.2.

|   | Simulation pair                                             | Resolution | Length    | Nudging | Notes                                     |
|---|-------------------------------------------------------------|------------|-----------|---------|-------------------------------------------|
| 1 | Reference 100 km; $N_c \ge 20 \text{ cm}^{-3}$              | 100 km     | 31 years  | No      | Create SPA files                          |
| 2 | SCREAM v1 12 km                                             | 12 km      | 13 months | Yes     | vs. 3: resolution sensitivity             |
| 3 | SCREAM v1 3 km                                              | 3 km       | 13 months | Yes     | SCREAM v1 ERFaer                          |
| 4 | E3SM-MAM v3; $N_c \ge 20 \text{ cm}^{-3}$                   | 100 km     | 13 months | Yes     | vs. 1: nudging sensitivity                |
| 5 | E3SM-MAM v3                                                 | 100 km     | 13 months | Yes     | vs. 4: N <sub>c</sub> limiter sensitivity |
| 6 | E3SM-SPA v3; $\alpha = 2000 (N_{\rm ccn} C_{\rm f})^{0.55}$ | 100 km     | 13 months | Yes     | vs. 7: activation sensitivity             |
| 7 | E3SM-SPA v3; $\alpha = N_{\rm ccn}C_{\rm f}$                | 100 km     | 13 months | Yes     | vs. 3: SPA-model sensitivity              |

The only cases with a  $N_c$  limiter are 1 and 4; all others have no limiter. Note that the  $N_c$  limiter is applied in-cloud, that is, in-cloud  $N_c \ge 20 \text{ cm}^{-3}$ . All other quantities throughout the manuscript are grid-mean values, unless explicitly noted otherwise.

to control for the large-scale circulation and to enhance the signal-to-noise ratio in the ERFaer calculations (e.g., Kooperman et al., 2012; Zhang et al., 2014; Forster et al., 2016; Zhang et al., 2022b); as long as the same nudging data is used for the present-day and pre-industrial runs, the ERFaer signal will be consistent regardless of the year or the reanalysis data used. In E3SM v3 testing we conducted, but not shown here, we found that systematic uncertainties in ERFaer due to choice of nudging data and period are approximately 0.1 W m<sup>-2</sup>. Additionally, in all nudged runs, the effects of aerosol perturbation on large-scale wind circulation are suppressed by design.

# 3 Results



### 3.1 SCREAM v1 ERFaer

The central question of this study is: Can the high-resolution SCREAM v1 configuration reproduce an aerosol effective radiative forcing (ERFaer) from pre-industrial to present-day conditions based on the SPA prescription from a low-resolution model that is similar to that produced by low-resolution models with fully interactive aerosols? In Figures 2 and 3, we show the time series and spatial distribution of the SCREAM v1 ERFaer, which is defined as the difference in the top-of-model radiative imbalance between the present-day and pre-industrial simulations (Forster et al., 2016). We assume all grid points provide robust signals; thus, we include information from all grid points in our analyses. It is immediately clear that SCREAM v1 does not reproduce the ERFaer from the reference E3SM v3 simulations; instead, it produces a significantly more negative ERFaer value, even though aerosols in SCREAM are prescribed based on the E3SM v3 simulations. Both horizontal resolutions of SCREAM v1 produce similar seasonal cycles and total numbers (Figure 2), but with slightly differing spatial distributions (Figure 3). Compared to E3SM v3, the SCREAM v1 configuration produces a more accentuated ERFaer signal spatially (Figure 3), resulting in a more negative global ERFaer.

**Figure 2.** Seasonal cycle of area-weighted globally averaged aerosol effective radiative forcing (ERFaer) from the SCREAM v1 configuration and the reference from which the SPA files were obtained (spatiotemporal average in the legend). The climatological reference SPA values are obtained from a simulation run at a 100-km horizontal resolution with E3SM v3 for 31 years with only the atmosphere and land components interactive without nudging (the first year is discarded). The SCREAM v1 configuration is run at two horizontal grid spacings (3 km and 12 km) forced with the same SPA files with other boundary conditions, while nudged to MERRA-2 data from August 1, 2019 to September 1, 2020; the first month (August 2019) is discarded.

While the ERFaer resolution sensitivity of the SCREAM v1 configuration appears to be minimal, the 12-km configuration produces a slightly more negative ERFaer signal than the 3-km configuration. At first glance, this may be surprising because of the significant differences in the base cloud and climate states between the two resolutions. The summary in Table 2 shows that while the base states at present-day settings are different, the perturbation from pre-industrial to present-day settings are broadly similar. The cloud droplet concentration at cloud top is similar, showing that SPA is effectively translating the cloud condensation nuclei prescription into cloud droplets consistently across resolutions, as designed in Equation 1. Additionally, nudging is employed in these studies to control for confounding factors that may distort the ERFaer signal. Specifically, the nudging is strong enough to keep large-scale features roughly constant across the perturbation (e.g., Gettelman et al., 2020).

Yet, we note that in the 12-km simulation shown in Figure 3, the more negative ERFaer signal appears in convectively active land regions (Amazon, East Africa, South Asia, Maritime Continent) where aerosol perturbations exist and/or orographic precipitation is common (Maritime Continent, Himalayas, Western Ghats, Ethiopian Highlands, tropical Andes). This potentially indicates that explicit convection representation and topographic resolution modulate regional ERFaer through their effects on clouds and precipitation. On the other hand, the reverse appears to be true in the marine stratocumulus regions west of conti-

**Figure 3.** Same as Figure 2, but showing a spatial distribution of the temporally averaged ERFaer (spatiotemporal average at top left). Note that here and elsewhere, all data are interpolated onto the coarsest unstructured output grid (150 km) for visualization.

nental masses that are so often the focus of aerosol–cloud interactions studies. In those regions, the ERFaer signal appears to be broadly more negative in the 3-km run than the 12-km run. Together, the opposing trends in the two regions suggest a globally insensitive ERFaer signal, but with a regional dependence on the resolution. Such details can help guide future research efforts to probe the regional aspect of aerosol–cloud interactions in high-resolution models.



We assess that the ERFaer signal is mostly due to aerosol-cloud interactions (indirect effects; approximately 90 % of the total ERFaer signal). Aerosol-radiation interactions (direct effects) are relatively small in the SCREAM v1 configuration, as they are relatively small in the reference simulations, compared to the indirect effects. The most significant contributions to the ERFaer signal are from the instantaneous radiative forcing due to changes in cloud droplet concentration (Twomey effect) and the adjustment of cloud content to the cloud droplet changes (cloud content adjustments). Through partial radiative perturbation

**Table 2.** Summary of area-weighted, globally averaged quantities in present-day (PD) conditions and the PD–PI differences in parentheses  $(\Delta)$ .

| SCREAM v1 quantity                       | 12 km PD (Δ)   | 3 km PD (Δ)    |
|------------------------------------------|----------------|----------------|
| TOA $\Delta F$ (W m <sup>-2</sup> )      | -2.06 (-1.95)  | 3.23 (-1.89)   |
| SW CRE (W $m^{-2}$ )                     | -51.18 (-1.20) | -46.29 (-0.98) |
| LW CRE (W $m^{-2}$ )                     | 26.00 (-0.22)  | 25.30 (-0.22)  |
| Cloud-top liquid $C_{\mathrm{f}}$        | 0.14 (0.00)    | 0.13 (0.00)    |
| Cloud-top ice $C_{\rm f}$                | 0.78 (-0.00)   | 0.79 (-0.01)   |
| In-cloud LWP (g $m^{-2}$ )               | 31.89 (2.48)   | 26.51 (1.45)   |
| In-cloud IWP $(g m^{-2})$                | 7.30 (0.17)    | 10.70 (-0.09)  |
| In-cloud $N_{\rm c,top}~({\rm cm}^{-3})$ | 64.49 (20.41)  | 65.10 (19.63)  |
| Cloud-top $R_c$ (µm)                     | 1.11 (-0.08)   | 0.98 (-0.04)   |
| Cloud-top $R_i$ ( $\mu$ m)               | 10.90 (-0.03)  | 12.38 (-0.07)  |

TOA  $\Delta F$  is the top-of-atmosphere net radiation imbalance. CRE is cloud radiative effects; values for shortwave (SW) and longwave (LW) are shown. Cloud-top quantities were calculated using the cloud-top statistical definition (Tiedtke et al., 1979, and references therein). Liquid water path (LWP) and ice water path (IWP) are calculated as the vertically integrated mass of liquid and ice water, respectively. The effective cloud droplet radius  $R_{\rm c}$  and effective ice crystal radius  $R_{\rm i}$  are shown at cloud top.

studies (Mülmenstädt et al., 2019), we find that the liquid and ice content adjustment terms account only for about 15% (-0.23 and -0.05 W m<sup>-2</sup> for liquid and ice water path adjustments, respectively) of the total ERFaer signal in the 12-km simulations, broadly consistent with previous studies (e.g., Zelinka et al., 2023, Figure 9, "cloud amount" component). Taken together, these results indicate that the Twomey effect is the most important component of the ERFaer signal in the SCREAM v1 configuration, which depends strongly on the aerosol activation process.

The default SPA translation of aerosol activation (Equation 1,  $\alpha = N_{\rm ccn}C_{\rm f}$ ) shows a significant overestimation in the indirect effect signal going from the reference E3SM v3 simulation to SCREAM v1 simulations. There are several potential explanations for the discrepancy. First, the models use different macrophysics schemes; second, they use different tunings of the same cloud microphysics scheme; third, one model has explicit convection while the other parameterized; fourth, one model uses a fully interactive aerosol scheme while the other completely bypasses that with a simple prescribed one. In the following sections, we show that the two most important factors are the tuning of the cloud droplet number concentration limiter employed in the E3SM v3 cloud microphysics scheme and the nature of the aerosol activation from cloud condensation nuclei to cloud droplets in SCREAM v1. To do so, we use the relatively inexpensive E3SM v3 (Section 2.1.2) to conduct additional sensitivity studies in Section 3.3.

# 245 3.2 Droplet activation





Because pure-water cloud droplets require an atmospherically infeasible supersaturation to form, cloud droplets must form on cloud condensation nuclei (CCN) that activate at a given supersaturation (Twomey, 1991, Section 5). Representation of the so-called CCN spectrum varies from simplified power-law functions (e.g., Twomey, 1959) to more complex mappings from aerosol size space to the supersaturation space (e.g., Nenes and Seinfeld, 2003). In the supersaturation space, it is possible to calculate  $N_{\rm ccn}$  from aerosol information at any supersaturation (Ghan et al., 2011, Equation 20). The activation of  $N_{\rm ccn}$  into  $N_{\rm c}$  depends on the underlying aerosol size distribution and its composition as well as atmospheric dynamics, especially updraft velocity, thus resulting in distinct regimes of activation (Reutter et al., 2009). While simplified activation schemes (e.g., Seifert and Beheng, 2006, Equation 17) and more complex activation schemes (e.g., Abdul-Razzak and Ghan, 2000; Ming et al., 2006; Morales Betancourt and Nenes, 2014) are routinely used, the relationship between  $N_{\rm ccn}$  and  $N_{\rm c}$  predicted by activation schemes remains variable and uncertain from a globally aggregated perspective (Gryspeerdt et al., 2023), despite the models' general agreement with more accurate but computationally expensive parcel models (e.g., Ghan et al., 2011; Rothenberg et al., 2018, and references therein). In summary, while theoretical studies along with laboratory and field experiments have improved our understanding of the aerosol activation process, uncertainties persist in its their remotely sensed and globally modeled impact on cloud droplet concentrations (McCoy et al., 2017; Hasekamp et al., 2019; Gryspeerdt et al., 2023).

Despite the uncertainty, it is thought that the relationship between  $N_{\rm ccn}$  (at constant supersaturation) and  $N_{\rm c}$  exhibits a loglog slope that is less than 1, specifically,  $0.3 \le d \ln N_c / d \ln N_{\rm ccn} \le 0.8$  (McCoy et al., 2017; Bellouin et al., 2020). This resulting sublinear relationship is often understood in the context of supersaturation variability and the transition from aerosol-limited to updraft-limited activation regimes. In order to better constrain  $\alpha(N_{\rm ccn}C_{\rm f})$  in Equation 1, we examine the correlations between these two variables in an E3SM v3 run in Figure 4. We choose to construct a correlation between  $N_{\rm ccn}C_{\rm f}$  and  $N_{\rm c}$  as they appear in Equation 1 using the present-day simulation of experiment 5 in Table 1, which does not include the  $N_c$  limiter. We note that E3SM v2 and v3 runs tend to have a frequent occurrence of low-N<sub>c</sub> values (Shan et al., 2024; Wan et al., 2025), which may distort the correlation; we thus filter by two different cutoffs in Figure 4 by discarding data below a given value. The first cutoff is at  $10^{-5}$  kg<sup>-1</sup> to avoid any unrealistic and spurious points at very low concentrations; the second cutoff is at  $10^6$  kg<sup>-1</sup> to avoid any bias introduced by the problematic frequent occurrence of low- $N_c$  values. The  $d \ln N_c / d \ln N_{ccn}$  slope of 0.55 is inspired by observational studies (Hasekamp et al., 2019; Gryspeerdt et al., 2023) and lies in the middle of the aforementioned assessed range (McCoy et al., 2017; Bellouin et al., 2020), but it generally varies depending on atmospheric conditions and underlying aerosol information. Based on the results in Figure 4, we estimate that  $\alpha = 2000 \, (N_{\rm ccn} C_{\rm f})^{0.55}$  likely summarizes the relationship in E3SM v3 without the potentially too-frequent low-N<sub>c</sub> values; as such, we use the new formulation in the sensitivity test in Section 3.3. A more advanced formulation could involve a piecewise function that captures the changing gradients of the aforementioned activation regimes more effectively, but we leave that for future work. We acknowledge the crude nature of this approach due to the confounding factors across space and time when it comes to the relationship between  $N_c$  and  $N_{ccn}$  (e.g., Andreae, 2009; Chen et al., 2011; Moore et al., 2013; Gryspeerdt et al., 2023; Varble et al., 2023; Ghosh et al., 2024).

Figure 4. Relationship between grid-mean  $N_{\rm ccn}C_{\rm f}$  and grid-mean  $N_{\rm c}$  in E3SM-MAM v3 model without the  $N_{\rm c}$  limiter. The  $N_{\rm ccn}$  shown here is the cloud condensation nuclei concentration at 0.10 % supersaturation. Because E3SM v3 tends to produce a high frequency of low- $N_{\rm c}$  values, we filter the data by two different cutoffs in  $N_{\rm c}$ :  $10^{-5}$  and  $10^6$  kg<sup>-1</sup>. The circles represent 150 logarithmic bins of the data obtained from 75 randomly selected temporal samples fully resolved in space and height. The count frequency in the legend is the number of data points in each bin, with the total bins being 129600000 (75 samples, 21600 horizontal columns, and 80 vertical levels). The solid lines are denoted by the equations in the legend. The  $d \ln N_{\rm c}/d \ln N_{\rm ccn}$  slope of 0.55 is inspired by observational studies cited in the main text, while the prefactor (2000) is chosen to best fit the data with the higher cutoff.

# 3.3 E3SM-SPA v3 ERFaer




In Figures 5 and 6, we show the results from the sensitivity tests conducted using the E3SM v3 configurations described in Section 2.2.2. As a confirmation, we performed a one-year nudged version (Figure 5) of the reference 31-year simulation (Figure 2), and we find a good agreement between the ERFaer signal from the reference and the one-year simulation. We find that the  $N_c$  limiter (in-cloud  $N_c \ge 20$  cm<sup>-3</sup>) has an effect of about a global average of 0.40 W m<sup>-2</sup> on the ERFaer signal compared to the case where it is not applied. The remaining difference between the reference simulation and the E3SM-SPA v3 simulation (about 1 W m<sup>-2</sup>) is due to the activation process in the SPA scheme. We find that we are able to roughly reproduce the SCREAM v1 ERFaer signal using the E3SM-SPA v3 configuration, with a global average of -1.89 W m<sup>-2</sup> in the SCREAM v1 3-km configuration, -1.95 W m<sup>-2</sup> in the SCREAM v1 12-km configuration, and -2.12 W m<sup>-2</sup> in the E3SM-SPA v3 100-km configuration. In Table 3, we show the results of further sensitivity tests to confirm the robustness of the ERFaer signal to the log-log slope of the relationship between  $N_{ccn}$  and  $N_c$  in the E3SM-SPA v3 configuration, showing a consistent trend of increasing ERFaer magnitude with increasing slope. Most importantly, we are able to almost exactly reproduce the global

Figure 5. Like Figure 2, but for the nudged E3SM v3 configurations tested.



ERFaer signal of E3SM-MAM v3 with the modified activation relationship in the E3SM-SPA v3 configuration using the fit from Figure 4.

In Figure 6, we note the modified activation parameterization ( $\alpha = 2000 (N_{\rm ccn} C_{\rm f})^{0.55}$ ) in the E3SM-SPA v3 configuration produces a broadly similar spatial distribution of the ERFaer signal compared to the interactive and more complex activation scheme in the E3SM-MAM v3 configuration. Nonetheless, there appear to be subtle differences between the two signals in several regions. The E3SM-MAM configuration appears to produce a more accentuated signal (positive and negative) than the E3SM-SPA configuration, though the overall global average is similar. This could be due to the deviations from the fit line in Figure 4 which effectively balances deviations above and below such that the net global result is similar. To investigate further, we examine the cloud droplet number concentration at cloud top in Figure 7. We find that the activation formulation in the E3SM-SPA v3 configuration has a significant effect on the cloud droplet number concentration at cloud top. Going from  $\alpha = N_{\rm ccn}C_{\rm f}$  to  $\alpha = 2000 \left(N_{\rm ccn}C_{\rm f}\right)^{0.55}$  we see an approximate decrease of 40 % in the present-day simulations (Figure 7 left) and an approximate decrease of 66 % in the PD-PI differences (Figure 7 right). Moreover, the revised activation formulation with the slope of 0.55 does not exactly match the more complex interactive MAM implementation, with a slight underestimation of cloud-top  $N_c$ , likely explainable by the fact that the data averages lie above the line corresponding to the slope of 0.55 in Figure 4. The fact that the globally averaged ERFaer signal calculated from experiments 5 and 6 in Table 1 are quite similar, but the cloud-top  $N_c$  values are different, suggests that the end climatological state may be achieved through different means. For example, it is plausible that the nature of the Twomey effect and adjustments to it may be fundamentally changed between the two configurations. We leave a more thorough investigation of these process pathways to future work.

Figure 6. Like Figure 3, but for the nudged E3SM v3 configurations tested.

# 4 Conclusions

Our main conclusion is that the global aerosol effective radiative forcing (ERFaer) can be constrained across different resolutions and configurations using a simple prescribed aerosol scheme. We find that SCREAM v1 ERFaer is dominated by

**Figure 7.** Like Figure 6, but showing the present-day values (left) and PD–PI differences (right) in grid-mean cloud-top  $N_c$ . We emphasize that the values represented are grid-mean values, which are often smaller than their in-cloud counterparts.

indirect cloud effects, principally through the Twomey effect from aerosol-induced cloud droplet number perturbations. This radiative forcing term (Twomey effect) is sensitive to aerosol activation assumptions, while global-scale cloud adjustments to

**Table 3.** The sensitivity of ERFaer to the log-log slope of the relationship between  $N_{\rm ccn}$  and  $N_{\rm c}$  in the E3SM-SPA v3 configuration while controlling for the prefactor. The slope is chosen first, and then the prefactor is modified to ensure all log-log fit lines approximately pass through the intersection of the two lines in Figure 4. These experiments are not reported in Table 1; they follow experiment 6 exactly but with different values of slope and prefactor in the activation equation.

| SPA activation                                       | ERFaer (W m <sup>-2</sup> ) |
|------------------------------------------------------|-----------------------------|
| $\alpha = (N_{\rm ccn}C_{\rm f})^{0.50} \times 4654$ | -1.03                       |
| $\alpha = (N_{\rm ccn}C_{\rm f})^{0.55} \times 2000$ | -1.16                       |
| $\alpha = (N_{\rm ccn}C_{\rm f})^{0.60} \times 860$  | -1.25                       |
| $\alpha = (N_{\rm ccn}C_{\rm f})^{0.80} \times 30$   | -1.70                       |
| $\alpha = (N_{\rm ccn}C_{\rm f})^{1.00} \times 1$    | -2.12                       |

the Twomey effect remain secondary. Consequently, we hypothesize that global ERFaer resolution dependencies may emerge primarily from how models resolve aerosol lifecycle processes and activation physics in aerosol schemes.





The default SPA linear activation formulation in our study using E3SM v3  $N_{\rm ccn}$  overestimates cloud droplet sensitivity to aerosol perturbations. Using a simplified activation formulation that inaccurately translates aerosol effects is not unprecedented. For example, the MACv2-SP scheme as originally formulated (Stevens et al., 2017, Figure 8 top) underestimated the cloud droplet sensitivity to aerosol perturbation (Herbert et al., 2021, Figure 3 c). As a result, subsequent studies adopted a stronger aerosol sensitivity formulation (Herbert et al., 2024). We modified the prescribed relationship between  $N_c$  and  $N_{\rm ccn}$ , which resulted in agreement of global ERFaer between E3SM v3 and SCREAM v1 across resolutions. Thus, this modified SPA formulation will be used as the default going forward. This adaptability constitutes a fundamental strength of prescribed schemes, enabling targeted hypothesis testing through controlled decoupling of aerosol–cloud interactions (Fiedler et al., 2017).

By holding the aerosol state (and thus forcing) to a preferred reference state, our framework enables different types of applications and studies. First, it enables studies of extreme weather and climate feedbacks independent from aerosol forcing uncertainty by imposing a targeted ERFaer. Second, it allows for the high-resolution investigation of cloud responses to aerosol perturbations without cloud-state feedback complications. The scheme's one-way coupling eliminates interactive feedback challenges like wet removal parameterization uncertainties (McCoy et al., 2020; Shan et al., 2024), allowing modular analysis of aerosol–cloud interactions. This proves particularly valuable for disentangling explicitly parameterized aerosol–cloud interactions mechanisms (e.g., precipitation suppression) from implicitly represented aerosol–cloud interactions mechanisms that arise from the interplay of multiple processes like entrainment feedbacks — the interplay in these process pathways remains critical for understanding aerosol-cloud interactions (e.g., Mahfouz et al., 2024; Mülmenstädt et al., 2024b, a). Finally, our framework opens the door for systematic identification of ERFaer deviation hot spots between simple (SPA) and complex (MAM) schemes for process-level scrutiny. Ultimately, the "true" ERFaer signal is not well constrained, and uncertainties persist in how close global circulation models are to representing it faithfully. The iterative approach of using a simple scheme

to constrain the signal and then using that signal to better understand and improve complex schemes is a promising avenue for future research.

While SPA currently utilizes E3SM v3 data, which has its own sets of biases (e.g., Xie et al., 2025), future implementations could integrate observationally constrained datasets like MACv2-SP (Stevens et al., 2017) while preserving aerosol spatial heterogeneity (Hassan et al., 2024). Further, the documented challenges in constraining the relationship between  $N_{\rm ccn}$  and  $N_{\rm c}$  (e.g., Gryspeerdt et al., 2023) indicate the need for a more advanced droplet activation treatment that takes into account aerosol information as well as atmospheric factors and cloud dynamics (e.g., Ghan et al., 2011). In the future, the SPA implementation could include a dynamic set of  $N_{\rm ccn}$  tracers with climatological relaxation to better capture the effects of large-scale circulation, macrophysics sub-grid variability, and cloud microphysics on aerosols. Such enhancements may bring SPA closer to the MAM implementation, while still retaining the advantages of efficiency and interpretability.






While SPA has advantages in efficiency, interpretability, and simplicity, the soon-to-be-available MAM implementation into SCREAM will explicitly predict the aerosol lifescyle, and thus likely offer a far superior representation aerosol-cloud interactions. The one-way coupling of SPA with clouds allows for easier access to certain process pathways, but the two-way coupling nature of MAM allows for a more realistic representation of aerosol effects. Moreover, the promise of high-resolution models like SCREAM to improve predictions relative to traditional, coarser-resolution models lies to a substantial extent in their ability to represent convection explicitly, and it is of keen interest to know how explicitly represented convection interacts with aerosols. In its current form, SPA hardly captures any of the links between convection and aerosols, such as the effects of variable updraft velocity on activation, efficient wet removal of aerosols, and vertical transport of aerosols as well as their precursors.

In summary, while constraining the global ERFaer signal is an important first step, it is only the beginning of fully understanding global aerosol–cloud interactions at higher resolution in SCREAM v1. The ability to constrain global states allows researchers to proceed with confidence in adjacent studies, such as those examining extreme weather events or climate feedbacks. However, focusing solely on global states ignores important regional details and may obscure important process-level insights, which remain crucial for comprehensive understanding. Having reproduced known ERFaer results with a global convection-permitting model opens opportunities to leverage these models for rigorous process-level and regional studies through carefully designed experiments targeting specific aerosol–cloud interaction pathways. It also sets the stage for a detailed comparison against the soon-to-be-available interactive aerosol scheme for the SCREAM configuration, as well as an iterative process to improve both the prescribed and interactive schemes.

Code and data availability. The default E3SM v3 code is publicly available at https://github.com/E3SM-Project/E3SM (specifically commit d9553ab is used in this study; last access: 30 December 2024). The SCREAM v1 code is available at https://github.com/E3SM-Project/scream (specifically commit 29bdb81 is used in this study; last access: 30 December 2024). We additionally provide a Zenodo archival repository with the following: (1) the SCREAM v1 code; (2) the default E3SM v3 code for E3SM-MAM runs; (3) the modified E3SM v3 code for E3SM-SPA runs; (4) the code to reproduce the plots in the manuscripts; (5) instructions to reproduce the runs in the manuscript; (6)

instructions to retrieve the data files produced by the simulations in this study, which have been publicly long-term archived using NERSC HPSS. The archival repository is available at https://doi.org/10.5281/zenodo.17069992 (Mahfouz, 2025).

Author contributions. All authors contributed to the study design, model development, model runs, data analysis, and/or manuscript writing.

Competing interests. At least one of the (co-)authors is a member of the editorial board of Atmospheric Chemistry and Physics.






Acknowledgements. This research was supported as part of the Energy Exascale Earth System Model (E3SM) project, funded by the U.S. Department of Energy (DOE), Office of Science, Office of Biological and Environmental Research. The Pacific Northwest National Laboratory (PNNL) is operated for DOE by the Battelle Memorial Institute under contract no. DE-AC06-76RLO1830. The data (E3SM v3 simulations) were produced using a high-performance computing cluster provided by the DOE Office of Science and Office of Biological and Environmental Research Earth System Model Development program area of the Earth and Environmental System Modeling program and were operated by the Laboratory Computing Resource Center at Argonne National Laboratory. This research (SCREAM v1 simulations) used resources of the National Energy Research Scientific Computing Center (NERSC), a U.S. Department of Energy Office of Science user facility located at Lawrence Berkeley National Laboratory, operated under contract no. DE-AC02-05CH11231 using NERSC award BER-ERCAP-0027116. Sandia National Laboratories is a multi-mission laboratory managed and operated by National Technology & Engineering Solutions of Sandia, LLC (NTESS), a wholly owned subsidiary of Honeywell International Inc., for the U.S. Department of Energy's National Nuclear Security Administration (DOE/NNSA) under contract DE-NA0003525. This written work is authored by an employee of NTESS. The employee, not NTESS, owns the right, title and interest in and to the written work and is responsible for its contents. Any subjective views or opinions that might be expressed in the written work do not necessarily represent the views of the U.S. Government. The publisher acknowledges that the U.S. Government retains a non-exclusive, paid-up, irrevocable, world-wide license to publish or reproduce the published form of this written work or allow others to do so, for U.S. Government purposes. The DOE will provide public access to results of federally sponsored research in accordance with the DOE Public Access Plan. This paper describes objective technical results and analysis. Any subjective views or opinions that might be expressed in the paper do not necessarily represent the views of the U.S. Department of Energy or the United States Government. This paper was performed under the auspices of the U.S. Department of Energy by Lawrence Livermore National Laboratory under Contract DE-AC52-07NA27344.

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
