# Peer review of "Prescribing the aerosol effective radiative forcing in the Simple Cloud-Resolving E3SM Atmosphere Model v1"

_EGUsphere, 2025_

## Referee Comment (RC1)

**Review of manuscript egusphere-2025-1868**

**Summary**

The authors present an evaluation of the effective radiative forcing due to anthropogenic aerosol changes in the SCREAM model. The aerosols are prescribed with a simplified scheme and are coupled to the radiation and cloud microphysics. They perform high-resolution simulations with the SCREAM model and compare those with coarse-resolution simulations with the E3SM model. They find a more negative radiative forcing due to aerosol-cloud interactions in the high-resolution simulations. They show that this difference is linked to the limiter of the cloud droplet number concentration in E3SM and the activation of aerosols to cloud droplets in SCREAM. They conclude that high-resolution simulations with prescribed aerosols allow us to perform rigorous process-level studies under controlled conditions. The paper is well written, well structured, and insightful. There are two general comments and some specific comments.

**General comments**

- The authors should describe in more detail how the cloud droplet number is handled in their simulations. Please list all processes, i.e., sources and sinks, that act on the prognostic cloud droplet number. And please explain whether the liquid cloud fraction is binary or continuous and whether it is subject to a threshold.

- The authors should describe in more detail how the aerosols are prescribed in their simplified scheme. Please add information about the aerosol optical depth at 550 nm for the present-day and pre-industrial scenario. That would also make it easier to compare their scheme with the MACv2-SP scheme of Stevens et al. (2017).

**Specific comments**

- Line 2: If possible, please shorten or split that long term, i.e., Energy Exascale Earth System Model (E3SM) Simple Cloud-Resolving E3SM Atmosphere Model (SCREAM) v1 configuration.

- Line 75: Is there a similar condition for the liquid cloud fraction? Is the liquid cloud fraction binary as well, i.e., either 0 or 1?

- Line 96: Please provide information about the aerosol optical depth to allow a comparison with the MACv2-SP scheme of Stevens et al. (2017).

- Line 110: Please list all processes, i.e., sources and sinks, that act on the prognostic cloud droplet number.

- Line 133 - 137: Please clarify what chemical processes are prescribed.

- Line 141: Please provide information about the aerosol optical depth of the present-day and pre-industrial simulation.

- Line 182: Does that estimate take into account the potential impact of aerosols on large-scale dynamics?

- Table 1: What is the minimum cloud droplet number concentration in the other simulations, i.e., 1 to 3 and 4 to 7?

- Figure 3: It looks like $10^{-5}$ should be replaced with $10^5$.

- Line 253: Please explain the filtering method in more detail.

---

## Author Comment (AC1)

**Re: egusphere-2025-1868**

**August 2025**

We appreciate the Reviewers' time and effort in engaging this manuscript. Our response is provided in blue to the Reviewers' comments in black.

**1 Reviewer 1**

**1.1 Summary**

The authors present an evaluation of the effective radiative forcing due to anthropogenic aerosol changes in the SCREAM model. The aerosols are prescribed with a simplified scheme and are coupled to the radiation and cloud microphysics. They perform high-resolution simulations with the SCREAM model and compare those with coarse-resolution simulations with the E3SM model. They find a more negative radiative forcing due to aerosol-cloud interactions in the high-resolution simulations. They show that this difference is linked to the limiter of the cloud droplet number concentration in E3SM and the activation of aerosols to cloud droplets in SCREAM. They conclude that high-resolution simulations with prescribed aerosols allow us to perform rigorous process-level studies under controlled conditions. The paper is well written, well structured, and insightful. There are two general comments and some specific comments.

Many thanks and much appreciation to the Reviewer! We have edited the manuscript according to the feedback to improve the clarity and presentation.

**1.2 General comments**

1. The authors should describe in more detail how the cloud droplet number is handled in their simulations. Please list all processes, i.e., sources and sinks, that act on the prognostic cloud droplet number. And please explain whether the liquid cloud fraction is binary or continuous and whether it is subject to a threshold.

   The prognostic cloud droplet number is treated much like in the cloud microphysics (P3) implementation, with the one change being before processing it in the P3 run loop, it is reset by the Equation provided in the main text (and below).

   $$N_c = \max\left(N_c, \alpha\left(N_{ccn}C_f\right)\right) \tag{1}$$

   In other words, the simplified scheme really only acts on the cloud droplet number concentration by resetting to a climatological mean if it falls below that mean. All other P3 processes in our model are sink processes; they are autoconversion, accretion, sedimentation, riming, and freezing. Thus, the only source process is the one represented by Equation 1. It is imperative note that cloud properties are transported by the dynamics solver and the macrophysics parameterization, which could impact them directly and indirectly. The liquid cloud fraction is continuous and produced by the macrophysics scheme in the model (SHOC).

   **Action:** Starting line 110 in the file tracked changes, we modify the text to clarify that the single source of droplets is the simplified scheme, and list all sink processes in P3 (autoconversion, accretion, sedimentation, riming, and freezing; we use the word riming explicitly and we add freezing). On line 77 in the file tracked changes, we state that SHOC is responsible for setting a subgrid liquid cloud fraction.

2. The authors should describe in more detail how the aerosols are prescribed in their simplified scheme. Please add information about the aerosol optical depth at 550 nm for the present-day and pre-industrial scenario. That would also make it easier to compare their scheme with the MACv2-SP scheme of Stevens et al. (2017).

   There are two parts to prescribing the aerosols in the simplified scheme. Firstly, the simplified aerosol scheme impacts the cloud droplet number concentration, which is described in the response above (ensuring the cloud droplet number concentration does not fall below a climatological mean set by cloud condensation nuclei, which is obtained from a 30-year-long E3SM v3 atmosphere-only 100-km run; Equation 1). Secondly, the optical properties of aerosols are taken from the same 30-year-long E3SM v3 atmosphere-only 100-km run. These optical properties are the three-dimensionally resolved and band-resolved single scattering albedo, asymmetry parameter, as well as the shortwave and longwave optical depths. The E3SM v3 100-km configuration is detailed in other concurrent manuscripts (which we cite in the manuscript). We provide the vertically summed optical depth at 550 nm for

[Figure]

Figure 1: The aerosol optical depth at 550 nm for the present-day (left) and pre-industrial (right) scenarios, both compared against the MACv2 observations from 2005. We note that the anthropogenic AOD500 is roughly 0.03 (comparing PD against PI) which is in line with benchmarks.

both the present-day and pre-industrial scenarios as well as in comparison with the MACv2 observations (taken from year 2005) below.

**Action:** We include the AOD figure in the response here. Starting line 149 in the file tracked changes, we add a brief discussion about the aerosol optical depth in the reference simulations from the v3 model.

**1.3 Specific comments**

1. Line 2: If possible, please shorten or split that long term, i.e., Energy Exascale Earth System Model (E3SM) Simple Cloud-Resolving E3SM Atmosphere Model (SCREAM) v1 configuration.

   Done. Thanks!

2. Line 75: Is there a similar condition for the liquid cloud fraction? Is the liquid cloud fraction binary as well, i.e., either 0 or 1?

   No, the liquid fraction fraction is entirely controlled by the macrophysics scheme which provides a continuous fraction that includes sub-grid information. We now note that in the main text as well.

3. Line 96: Please provide information about the aerosol optical depth to allow a comparison with the MACv2-SP scheme of Stevens et al. (2017).

   See Figure 1 for the aerosol optical depth at 550 nm for both present-day and pre-industrial scenarios.

4. Line 110: Please list all processes, i.e., sources and sinks, that act on the prognostic cloud droplet number.

   Done.

5. Line 133 - 137: Please clarify what chemical processes are prescribed.

   We remove the word "chemistry" from this sentence. What we meant here was we *implicitly* prescribe aerosols and *their* chemistry. By prescribing the ultimate details of aerosols (optical properties and CCN), we are implicitly prescribing chemical processes leading to those such as how ozone contributes to sulfate production. Because this is a technical detail, and it is wholly implicit, we decided to remove the word "chemistry" to avoid confusion. Perhaps the best description of this is "aerosol effects," which we now use in the text.

6. Line 141: Please provide information about the aerosol optical depth of the present-day and pre-industrial simulation.

Done. See Figure 1 for the aerosol optical depth at 550 nm for both present-day and pre-industrial scenarios. We also added that the anthropogenic AOD is 0.03 to this section.

7. Line 182: Does that estimate take into account the potential impact of aerosols on large-scale dynamics?

   We strictly mean the systematic uncertainty due to the choice of nudging data and period. In all nudged runs, the effects of aerosol perturbation on large-scale wind circulation are suppressed.

8. Table 1: What is the minimum cloud droplet number concentration in the other simulations, i.e., 1 to 3 and 4 to 7?

   Only 1 and 4 have the limiter. We add this to the note at the bottom of the table. And we add the limiter to the first experiment as well.

9. Figure 3: It looks like $10^{-5}$ should be replaced with $10^5$.

   It is $10^{-5}$. This very low filter was applied to ensure we are not including any spurious points in the analysis. We add some discussion on this in the main text as well.

10. Line 253: Please explain the filtering method in more detail.

    We add that the filtering here means discarding values below the cutoffs.

**2 Reviewer 2**

**2.1 Summary**

Mahfouz and co-authors present a new simplified framework for representing aerosol effects on radiation and clouds (SPA; a simplified prescribed aerosol scheme) that maintains some degree of fidelity regardless of model spatial resolution. This is timely given the move towards kilometre-scale earth system models that complement the coarser resolution models traditionally used for CMIP-style experiments. Having a framework that is consistent across model configurations provides an important tool for quantifying the role of aerosols in the earth system.

In this study, the authors produce aerosol climatologies for pre-industrial and present-day climates using a coarse-scale ( 100s km) version of E3SMv3 that includes a detailed aerosol microphysics model. The climatologies are used to prescribe time-varying 3D fields of aerosol optical properties in the higher-resolution (10s km) version of E3SMv3, thus representing aerosol-radiation interactions associated with anthropogenic activity. For aerosol-cloud interactions, the authors couple the aerosol climatologies to the cloud microphysics scheme via a function that relates the aerosol concentration to a cloud droplet number concentration. Sensitivity tests are performed to find a setup of SPA that best reproduces the global ERFaero that the coarse-resolution version of E3SMv3 produces. The SPA scheme and its associated methodology is an excellent addition to the community's ability to represent aerosol processes in an idealized/prescribed framework.

I thoroughly enjoyed reading this manuscript, and believe it is well suited for publication in ACP. However, I have some minor comments I would like to be addressed before recommending publication.

Many thanks and much appreciation to the Reviewer! We have edited the manuscript according to the feedback to improve the clarity and presentation.

**2.2 General comments**

1. This study implicitly assumes that E3SM-v3 is accurately representing aerosol distributions (throughout the industrial era) and Nc distributions. Have these been evaluated? If these evaluations are yet to be performed and published it may undermine this study. Can the authors include these evaluations themselves? Given that Nc ends up being an important contributor to the sensitivity can the authors compare to the Nc observations from Grosvenor et al 2018? If these evaluations cannot be included/referenced I recommend the authors explicitly state these as potential caveats/limitations of the study.

   We do not mean to imply that E3SM v3 is perfect, but rather that it is the available model for this purpose within the E3SM project. In fact, we do note in the conclusion section that a future version of SPA can incorporate observational data directly instead of relying solely on E3SM v3 data. We also agree that a comparison to the data in Grosvenor et al. 2018 should be a priority when assessing the performance of E3SM v3 and a future version of SCREAM with interactive aerosols.

   **Action:** We add "We note that the choice of E3SM-MAM v3 as the reference data for SPA implies that relevant biases in the reference low-resolution model will be in the new high-resolution model" to the reference simulation section (line 152 in the file with tracked changes); we add "which has its own sets of biases" to the statement of SPA using E3SM v3 data in the conclusion (line 332 in the file with tracked changes).

2. It is recognised that an aerosol perturbation in one region may have non-local effects associated with large scale circulation. This in turn can impact subsidence rates, moisture transport etc and represent non-local aerosol feedback. I would expect the nudging of the U/V wind components to prevent, or at the least dampen, these effects. Do the authors recognise this as a limitation?

We do recognize the limitation. However, regional signals (especially, as the Reviewer points out, in comparing the 12- and 3-km runs) are evident in these runs. The goal of this study is strictly the global signal, which — for now — ignores detailed regional implications. We respond more about this in the following point, where the Reviewer rightly points out regional information as a primary benefit of higher-resolution models.

3. One of the primary benefits of using kilometre-scale configurations for examining aerosol processes is the opportunity to study regional-scale responses of the atmosphere/clouds/radiation to the aerosol perturbation. By focusing on the global ERFaero, the regional-scale fidelity is sacrificed. Is this appropriate? Would a region-to-region comparison (or series of limited domains) be a better suited test?

We agree with the Reviewer about the importance of regional-scale responses. In this manuscript, we mainly focus on the global scale to establish a baseline for further studies. Our underlying assumption is that it is critical that the global-scale response is well constrained and understood before process- and regional-scale studies can commence. After all, if the global signal isn't fully constrained (say with an ERFaer of $-5$ W m$^{-2}$ or other highly unrealistic responses), could we trust regional- and process-scale results? The challenge, of course, is to get all these scales right, and not end up with conflicting signals...

**Action:** We highlight this limitation in the last paragraph of the conclusion (line 352 in the file with tracked changes); and we point to regional- and process-level studies as future endeavors.

4. Throughout the manuscript, the authors discuss the robust ERFaer signal that is achieved through the nudging. Are all grids included in the global magnitudes or only those that provide a robust signal? If there is a lot of heterogeneity in the statistical significance of the grid points I suggest the authors include stippling/masking to demonstrate this in the global plots — otherwise include a note in the manuscript to make the reader aware that all grid points can be considered significant/robust etc.

We implicitly assume that all grid points provide robust signals. We believe a more rigorous approach may entail filtering grid points by statistical significance, but we do not do that here for simplicity and continuity with previous studies.

**Action:** We add "We assume all grid points provide robust signals; thus, we include information from all grid points in our analyses" at the start of the results section (line 196 in the file with tracked changes).

**2.3 Specific comments**

1. Line 3 "we quantify the forcing due to anthropogenic aerosol changes using a simplified prescribed aerosol scheme. . . ". I think it is more accurate to state that you are assessing the sensitivity of the forcing. In reality, the aerosol forcing that has already been quantified by E3SMv3 and is now being used to constrain SCREAM via a series of sensitivity experiments.

We agree; we adopt the Reviewer's suggestion in rephrasing this sentence.

2. Line 5. "Nudged simulations at 3 km and 12 km horizontal grid spacings reveal a more negative aerosol forcing than the reference 100-km 5 E3SM v3.." This would imply that model-based estimates of the global aerosol forcing are dependent on resolution. This may well be the case, but I don't think this is supported by the study. The sensitivity experiments suggest that there is considerable impact from a combination of the Nc limiter in the reference simulation and an inadequate representation of ACI that (initially) fails to capture the coarser-scale configuration. Between these two effects it is impossible to say there is resolution-dependence between the 100 km and 3/12 km configurations. I suggest this is rewritten to better capture the study outcomes.

We think this sentence is accurate on its own. The results from Figure 1 show a significantly more negative ERFaer in SCREAM runs compared to the reference run. We agree it may imply a resolution sensitivity, but that is quickly clarified not to be case the in subsequent sentences (we write "exhibits little overall resolution sensitivity" on line 6). Thus, we agree with the reviewer this is potentially misleading, but we think the immediate clarification in the next sentence is sufficient.

3. Line 6 ". . . exhibits little overall resolution sensitivity". As a global ERFaero I would agree, but there is much more sensitivity at the regional scale — as highlighted by the authors on line 210. I would argue that differences between these two configurations actually do demonstrate resolution dependence (unlike 100km vs 12km). The only difference between the 3 and 12 km configurations is the resolution — the ACI/ARI treatments in this case are identical and therefore a better comparison. At the least, I suggest that the authors expand this sentence to highlight the spatial differences between the 3 and 12km configurations.

We agree; we add "while hints of resolution sensitivity appear regionally between the 3-km and 12-km runs" to the sentence.

4. Line 75. Is this also an assumption explored in Caldwell et al 2021? If so, I recommend adding the reference at the end of this sentence for clarity.

  Yes, and done.

5. Line 80/106. How is Nc in the P3 scheme coupled to the radiation scheme? Do changes in Nc influence the droplet effective radius in RRTMGP?

  Yes. RRTMGP uses the droplet effective radius in its calculation. The droplet number and mass concentrations influence the drop size distribution and so they control the effective radius. (Note that RRTMGP also uses the droplet mass mixing ratio directly.)

6. Line 124. In SCREAM v1 all clouds (I presume) are represented by P3, therefore the SPA scheme can influence all clouds. Is this also the case in the coarse configuration of E3SMv3? Is the aerosol microphysics scheme coupled to cloud microphysics in deep convection? / all clouds? If not, then are you comparing like for like? Perhaps expand this to fully describe how aerosols influence different clouds in the E3SMv3 configuration.

  That is a good point. In E3SM v3, as mentioned in the main text, the deep convection treatment includes cloud microphysics (treated separately from P3). We edit the description around this line to read "... convection scheme that also includes cloud microphysics effects that are modulated by the interactive aerosols..." to highlight that the microphysics extension in the deep convection scheme takes into account aerosol information.

7. Line 172. Why do you choose to nudge all levels rather than only the top 70 as in Zhang et al. 2022a?

  There are generally two reasons for avoiding to nudge a few layers near the surface. The first one is to avoid strongly constraining natural emissions (dust, sea salt). The second one is to allow lower boundary layer processes to evolve more freely. We thought it would be best to nudge all levels because that would be the more consistent approach with all configurations we have in this study (interactive aerosols or not; higher and lower resolutions). Overall, we are somewhat confident that the ERFaer metrics discussed in this manuscript will not be impacted by this decision.

8. Line 174. Do you mean ERA5 instead of MERRA-2?

  All experiments were nudged towards MERRA-2 data. The point we are trying to make here is that the nudging period does not matter much (nor does the source of the data, even though we used the same source).

9. Line 176. Was 2009-2010 a similarly weak el-Nino year?

  Not to our knowledge. The main reason for using 2009–2010 is data availability for the lower-resolution model...

10. Table 1. For clarity, the authors could add Nc $\geq 20$ cm$^{-3}$ to the reference case. Also, the footnote should be moved to the methodology when defining/explaining the Nc limiter (see comment further down)

  We agree; done.

11. Line 195. Is the 12 km configuration able to sufficiently represent convection? I thought this was in the gray zone?

  You are right that it is in the gray zone. That's part of why we include it in this study. In a concurrent study (Terai et al. 2025) on cloud feedbacks (utilizing identical present-day baseline setup, but without nudging), the team found significant differences between the 12- and 3-km configurations. We wanted to see if the same resolution sensitivity could be seen in the ERFaer as well...

12. Line 218. Typo respecticly

  Fixed. Thanks!

13. Line 251. Table 2 should be Table 1?

  Yes. Fixed. Thanks!

14. Line 251. The $N_c$ limiter. My understanding is that the standard E3SMv3 configuration uses a $N_c \geq 20$ cm$^{-3}$ limiter — is this correct? This isn't defined/explained in the methodology — please include or make clear. It took me a while to (hopefully correctly) realise that this is the default setting.

  We apologize for the confusion. You are right that the $N_c$ limiter is default in E3SM v3. This important piece of information is buried in the sentence "Similar to SCREAM v1, it uses the P3 scheme for stratiform cloud microphysics parameterization..., but with slightly different tuning parameters..." which now we explicitly state.

15. Line 252. Are the frequent occurrence of low $N_c$ values anonymously low with respect to observations? I presume this is why the default reference configuration limits $N_c$. If this is the case, then why do the authors work towards reproducing the global ERFaero from the non $N_c$-limited configuration?

[Figure]

Figure 2: Like Figure 4 in the main text, but including the "reference" line from Figure 1 in the main text.

There are two types of low-$N_c$ values. There are some that are super unrealistically low (say below $10^{-5}$ kg$^{-1}$) and there are ones that are realistically low (say slightly below $10^6$ kg$^{-1}$), but occurring in high enough frequency to cause noticeable changes. That is what inspired us to use two different cutoffs in Figure 3. While it benefits some metrics (like reducing the magnitude of ERFaer), we believe the the $N_c$ limiter is artificial and hard to justify. We note that we could alter $\alpha$ formulation in Equation 1 to instrument the effects of the limiter, but we decided not to do so.

16. Line 256. By fitting a linear (log–log) function the gradient of d$N_c$/d$N_{ccn}$ is constant — yet as the authors note this is not reflected by the E3SMv3 data. For the phase space that represent most of the data, the resulting gradient is too shallow (centre of the plot) — thereby dampening the sensitivity of $N_c$ to $N_{ccn}$. I wonder whether the best function is a three-regime function that would better capture the changing gradients. This wouldn't add much computational burden and would better reflect the variable gradients.

Yes, we concur. We do highlight the crude nature in the text (line 258 in submitted manuscript). We now add a more explicit sentence about a potential "upgrade" of the single-piece relationship used in the text, namely "A more advanced formulation could involve a piecewise function that captures the changing gradients more effectively, but we leave that for future work."

17. Line 263. Is this Section reference correct?

Thanks for spotting this mistake. Corrected.

18. Line 264. Please can you add the reference simulation to Figure 4 to demonstrate the good agreement?

It would not be a good comparison in this context. The reason is that the reference simulation (1 in Table 1) is a free-running one that spans 31+ years. A comparison between experiments 1 and 4 in Table 1 could tease out the nudging sensitivity, which is beyond the scope of this manuscript. We use experiment 4 as the nudged proxy for the reference, which is a decent proxy *measured annually*. See added Figure 2 in this response.

19. Line 291. Main conclusion. The aim is to reproduce E3SM-MAM with E3SM-SPA. This is achieved when comparing E3SM-MAM that is not limited by $N_c$ , with E3SM-SPA using the $N_{ccn}$ -$N_c$ function derived from E3SM-MAM data when limited by $N_c$ . Does this not point to a fundamental discrepancy? None of the sensitivity simulations are able to reproduce the E3SM-MAM reference ERFaero value of $-0.74$ W m$^{-2}$. Do you think this demonstrates an ability to constrain the model across different resolutions? I would say this is the only the case when $N_c$ is not limited — but then don't you have an unrealistic model configuration?

This is an excellent point/question. In short, yes, we think this demonstrates an ability to constrain the model across different resolutions. The $N_c$ limiter's purpose is to restrict how the microphysics scheme (P3) operates in the model. The SPA scheme sets a lower bound

on $N_\text{c}$ as prescribed by a meterological average of $N_\text{ccn}$. While the two mechanisms appear to act in the same way, they fundamentally do not. The latter (SPA) is much looser in its restrictions on the evolution of $N_\text{c}$. As shown in Table 3 in the submitted manuscript, we can tune the SPA formulation even further to get other ERFaer metrics; the fit in Figure 3 is not perfect, and there's room for improvement (including as the Reviewer points out, a piecewise function for $\alpha$ in Equation 1). And while the chosen fit from Figure 3 balances different regimes and ends up producing reasonably similar global answers, important regional and process details may have been overlooked. Ultimately, it depends on what the ultimate goal is, and we think the SPA scheme is effective at constraining global signals relatively well with justifiable tuning decisions; we do not make any significant claims beyond the global signals.